# Kendall Shape-VAE : Learning Shapes in a Generative Framework

**Sharvaree Vadgama**          S.P.VADGAMA@UVA.NL
*University of Amsterdam*

**Jakub Tomczak**          J.M.TOMCZAK@VU.NL
*Vrije Universiteit Amsterdam*

**Erik Bekkers**          E.J.BEKKERS@UVA.NL
*University of Amsterdam*

**Editors:** Sophia Sanborn, Christian Shewmake, Simone Azeglio, Arianna Di Bernardo, Nina Miolane

## Abstract

Learning an interpretable representation of data without supervision is an important precursor for the development of artificial intelligence. In this work, we introduce *Kendall Shape*-VAE, a novel Variational Autoencoder framework for learning shapes as it disentangles the latent space by compressing information to simpler geometric symbols. In *Kendall Shape*-VAE, we modify the Hyperspherical Variational Autoencoder such that it results in an exactly rotationally equivariant network using the notion of landmarks in the Kendall shape space. We show the exact equivariance of the model through experiments on rotated MNIST.

**Keywords:** generative models, variational autoencoder, geometry, Kendall shape, ideograms, equivariance, unsupervised learning

## 1. Introduction

The learning of reliable, compressed, and information-rich representations is essential in machine learning. Among the many unsupervised likelihood-based frameworks, Variational Autoencoders (Kingma and Welling, 2013; Rezende et al., 2014) have proven to be an intuitive framework with compelling results and have thus gained popularity in representation learning and generative image modeling. VAEs allow for learning of low dimensional latent representations, and can, e.g., be adapted for learning disentangled representations; (Chen et al., 2018) or modeling of dependencies between random variables in separated latent spaces (Ilse et al., 2020).

There has been considerable interest in making the latent space invariant to a transformation, but it has been tricky to do so with a VAE framework as it involves separate training of invariant latent spaces for specific transformations (Bepler et al., 2019) which cannot be generalized well. Adding a *geometric structure to latent space* has improved on the generation capacity of VAEs (Chadebec and Allassonnière, 2021), and could improve on the notion of disentanglement if the geometry is biased towards it (Davidson et al., 2018). Alternatively, a few works provide post-hoc explanations via classifiers that map the latent space to outcome, but they come with their limitations (Liu et al., 2019). For the tasks

of domain generalization (Ilse et al., 2020; Mitko, 2019) and compressed sensing (Kuzina et al., 2022) it is important to have explicit control over how the latent space is encoded.

We see a lot of potential in using geometry to structure VAE latent spaces as to obtain interpretable latents. Towards this end, we present a new VAE framework, the *Kendall Shape* VAE. It encodes each image as a geometric symbol, or *shape*, and does so in a rotation equivariant manner. Our approach is motivated by our daily use of symbols, pictograms, and ideograms in general, to represent concepts in a concise geometry form. As such, the learnt symbols in our Kendall Shape VAEs should be thought of as **neural ideograms**[1]. We illustrate the interpretability of our method by showing how similar the latents of the two different images of the same class look as well as show that the shape remains consistent.

Our contributions in this paper can be summarized as:

- We introduce Kendall Shape VAE, a fully *equivariant* VAE with a hyperspherical latent space, equivalent to a Kendall shape space.

- Points in latent space correspond to landmarks that define *shapes* in Kendall shape space. We show that equivalence classes of images are mapped to invariant shapes.

- We show that this *shape* is truly invariant to rotation and translation, by the use of equivariant encoder-decoder, and that our method *disentangles shape from pose*.

- We perform experiments in a simple unsupervised manner without any special equivariant loss or conditional alignment.

## 2. Related Work

Variational Autoencoders, as introduced by Kingma and Welling (2013) and Rezende et al. (2014), have a great advantage of fast and tractable sampling compared to other likelihood based frameworks. Since then a lot of work has been done to improve on VAE models by reducing the gap between approximate and true posterior distributions (Rezende and Mohamed, 2015; Kingma et al., 2016; Cremer et al., 2018), formulating tighter bounds (Li and Turner, 2016; Burgess et al., 2018b), for extending VAEs to discrete variables (Rolfe, 2017) and addressing posterior collapse (Lucas et al., 2019).

Hierarchical VAEs with advanced neural network architectures with multi-scale dependency between compressed representations, were presented in (Vahdat and Kautz, 2020; Child, 2020). In (Oord et al., 2017) the prior is learnt and hence results in higher quality of generation. Many works have focused on disentanglement of latent spaces (Skafte and Hauberg, 2019; Burgess et al., 2018a; Chen et al., 2018) and explaining latent space of VAEs (Liu et al., 2019).

Our work marries two classes of VAEs in a framework of Kendall shape space VAEs; *equivariant* and *hyperspherical*. Lafarge et al. (2020) proposes an equivariant VAE, where the latents are equivariant without having information of pose or orientation. Bepler et al. (2019) disentangles translation and rotation in an image by training a VAE on a spatial grid. Kuzina et al. (2022) introduces an equivariant prior and results in approximate rotational

---

1. Kendall Shape VAEs are not limited to inferring geometries of objects as you would with *pictograms* s.a. 🐦 for representing birds; more generally they represent concepts via *ideograms* s.a. ☺ for being happy.

equivariant latents. These equivariant methods have the usual perks of equivariance being an important inductive bias for image data that significantly reduces model complexity (Elesedy, 2022). They do, however, suffer from the same limitations as plain VAEs; namely, the latent space is Euclidean and the common Gaussian priors limits utilization of the space only locally around the origin. Hyperspherical VAEs (Davidson et al., 2018, 2019), on the contrary, allow for uniform priors on the sphere, which makes that the entire space (which is compact) is utilized. Our work inherits best of both equivariant and hyperspherical VAEs.

## 3. Background

As a preliminary to our method, we first review VAEs and Hyperspherical VAEs (SVAEs).

**VAEs**: Variational autoencoders consist of a generator $p_\theta(x|z)$, a prior $p_\theta(z)$ and an approximate posterior (encoder?) $q_\phi(z|x)$. When we parameterize the joint distribution by a neural network the marginalization over the latent variables (as to obtain $p_\theta(x)$) is untractable. So instead, our objective is to maximize the evidence lower bound (ELBO), using an approximate posterior $q_\phi(z|x)$. The ELBO is given by

$$\mathcal{L}(\theta, \phi) = \mathbb{E}_{z \sim q_\phi(z|x)} \log p_\theta(x|z) - D_{KL}[q_\phi(z|x)||p_\theta(z)].$$

**Hyperspherical VAEs**: In the original VAE, the prior and posterior are both defined as a normal distribution. In SVAE, however, we need to work with distributions on the sphere, for which the von Mises Fisher (vMF) distribution is the natural choice as it is the stationary distribution of a convection-diffusion process on the hypersphere $S^{m-1}$, just like the normal is on $\mathbb{R}^m$. The probability density function of the vMF distribution for a random unit vector $z \in S^{m-1}$ is defined as

$$q(z|\mu, \kappa) = \mathcal{C}_m(\kappa) \exp(\kappa \mu^T z),$$

where

$$\mathcal{C}_m(\kappa) = \frac{\kappa^{m/2-1}}{(2\pi)^{m/2}\mathcal{I}_{m/2-1}(\kappa)},$$

$||\mu||^2 = 1$, and $\mathcal{I}_\nu$ denotes a modified Bessel function of the first kind at order $\nu$. The Kullback–Leibler divergence can be analytically computed between a vMF distribution $vMF(z|\mu, \kappa)$ and a uniform distribution on a hypersphere $S^{m-1}$ $U(x)$ via

$$KL(vMF(\mu, \kappa)||U(S^{m-1})) = \kappa \frac{\mathcal{I}_{m/2}(k)}{\mathcal{I}_{m/2-1}(k)} + \log \mathcal{C}_m(\kappa) - \log \frac{2(\pi^{m/2})^{-1}}{\Gamma(m/2)}.$$

## 4. Kendall Shape VAE (KS-VAE) Framework

In (Davidson et al., 2018), the use of von Mises Fisher (vMF) distribution as a prior is motivated to support learning distributions lying on non-euclidean manifolds. Consider a case where the data lies on a circle; the normal VAE tries to force the embedding to be mapped into an approximated posterior distribution that has support on the entire $S^1$. But this fails, as in the low dimensions, Gaussian distribution presents probability mass around the origin, while in high dimensions, it tends to resemble a uniform distribution on the

surface of a hypersphere, with majority of its mass concentrated on the hyperspherical shell. By approximating the posterior using vMF in SVAE, the resulting latents are separated on the surface of the hypersphere and in addtition the model learns more efficiently. In this work we leverage the benefits of SVAEs, and additionally give a new interpretation by considering hyperspherical latent vectors as meaningful aligned set of points by relating them to landmarks in Kendall shape spaces. *This new viewpoint on hyperspherical VAEs further gives insight in how to make them equivariant to rotations*, as we show in the next section.

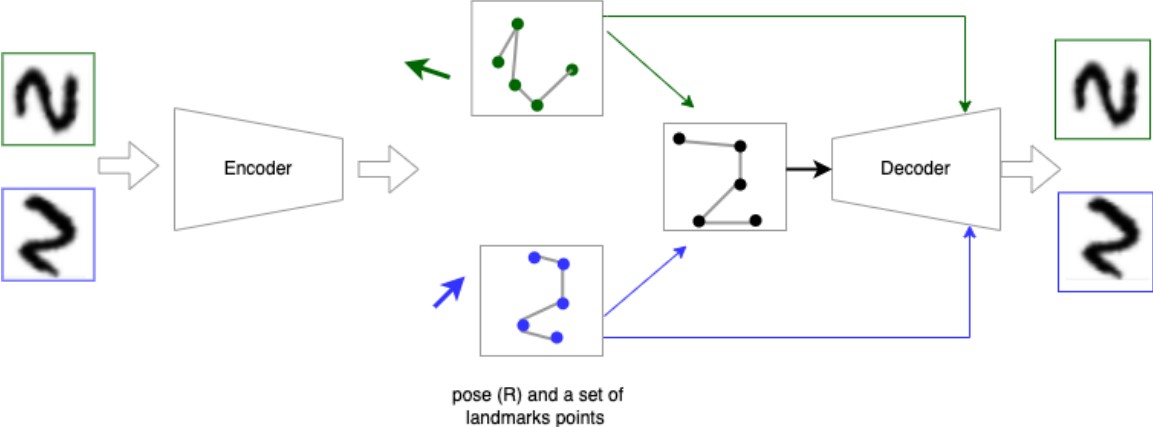

pose (R) and a set of
landmarks points

Figure 1: KS-VAE pipeline : Images pass through an equivariant encoder and give a pose, denoted by an arrow (in $R^2$ for simplicity) and a set of landmarks, denoted by dots. (We connect them with lines to show a *geometric symbol*). The encoder maps both of these image from the same class to a canonical orientation and a set of *landmarks*. This is then decoded and reconstructed using the extracted pose to give the final reconstructed images

## 4.1. Kendall Shape Spaces

(Kendall et al., 1999) defined pre-shapes by $k$ labelled points in Euclidean space $\mathbb{R}^m$. Two configurations of $k$ labelled points are regarded as the same shape if either of their pre-shapes can be transformed into the other by a rotation about a shared centroid. The quotient structure was extended in (Kendall et al., 2009) by defining shape space as the quotient of the space of landmark configurations by the group of translations, scale, and rotations.

We follow the notation of (Guigui et al., 2021) which defines a *pre-shape* space $S_m^k$ with $k$ points $\in \mathbb{R}^m$ as the space of $m \times k$ matrices, $M(m, k)$. We are interested in mapping the equivalence class formed by invariant transformations on this set of points or landmarks. To do so we remove the effects of translation (subtract the mean) and remove effects of

scaling (divide by Frobenius norm). This defines the pre-shape space

$$S_m^k = \{x \in M(m,k) | \sum_{i=1}^{k} x_i = 0, ||x|| = 1\}$$

which is identified with the hypersphere of dimension $S^{m(k-1)-1}$. To remove the effect of rotations, we define the equivalence relation $\sim$ on $S_m^k$ by $x \sim y \iff \exists \mathbf{R} \in SO(m)$ such that $y = \mathbf{R}x$ for all $x, y \in S_m^k$. For $x \in S_m^k$, let $[x]$ then denote its equivalence class for $\sim$. This equivalence relation results from the group action of $SO(m)$ on $\mathbb{R}^m$. From computational perspective, we need a vector, which we label as $x_0 \in S_m^k$, to represent the entire equivalent class $[x_0] = \{\mathbf{R}x_0 | \mathbf{R} \in SO(m)\}$. As a flattened vector $x_0 \in \mathbb{R}^{km}$, this vector transforms via the $SO(m)$ group representation $\rho(\mathbf{R}) := \oplus_k \mathbf{R}$, i.e., a block diagonal matrix with $\mathbf{R}$ along the diagonal.

## 4.2. The Kendall Shape VAEs Requires Equivariant Encoders/Decoders

In our KS-VAE setting, we consider mappings from (square integrable) images $f \in \mathbb{L}_2(\mathbb{R}^m)$ to a latent pre-shape representation $z \in S_m^k$ via an encoder Enc : $\mathbb{L}_2(\mathbb{R}^m) \to S_m^k$. *This mapping should be equivariant if we want the landmarks in $S_m^k$ to be geometrically meaningful* with respect to the content in the input image. That is, the content of the image may exhibit features or objects at a certain pose/orientation $\mathbf{R}$, like the rotation of a digit in the rotated-MNIST dataset, or the arbitrary rotation under which a cell might present itself under a microscope in histopathology (Lafarge et al., 2020). We would want that if the input image merely rotates that *the landmarks rotate accordingly* and the actual *shape remains invariant* (its content stays the same but only the orientation changes). As often we are interested in the invariant content of the image $f$, we can talk about an equivalence class of images $[f]$, where each image this class is related via a rotation.

It is important to understand that the equivalence classes themselves are invariant to scaling, translation and rotations, but the individual members $z \in [z]$ and $f \in [f]$ are covariant vectors under the action of $SO(m)$ and only invariant to translation and scaling. That is, an encoder Enc : $\mathbb{L}_2(\mathbb{R}^m) \to S_m^k$ should be equivariant via

$$\forall_{g=(\mathbf{x},\mathbf{R},s)\in SIM(m)}: \quad \text{Enc}(\mathcal{L}_g f) = \rho(\mathbf{R})\,\text{Enc}(f).$$

with $SIM(m)$ as a group of rotations, translations and scale on $SO(m)$ and $\mathcal{L}_g$ the left-regular representation of $SIM(m)$ on $m$-dimensional images, given by $\mathcal{L}_g f(x) = f(g^{-1}x)$. Similarly, we want the decoder Dec : $S_m^k \to \mathbb{L}_2(\mathbb{R}^m)$ to be equivariant. This can be ensured if we let them be parameterized by group equivariant CNNs (Cohen and Welling, 2016), specifically by $SIM(m)$-CNNs as proposed in (Knigge et al., 2022).

## 5. Implementation

### 5.1. Rotation Equivariant Layers

In this work we focus on the notion of rotation equivariance, and assume the datasets do not exhibit scale variations[2] such that our networks do not have to be scale invariant.

---

2. This is a reasonable assumption in many scenarios such as medical imaging where data is often acquired under a fixed resolution. Scale invariance can be guaranteed with methods such as (Knigge et al., 2022).

We then require $SE(m)$ equivariant neural networks. We rely on the `escnn` library (Cesa et al., 2022; Weiler and Cesa, 2019), which provide a general program for building $E(m)$ equivariant neural networks. It is based on the construction of feature fields with fibers (vector-valued feature channels) that transform under specific representations of $SO(m)$.

In light of the `escnn` library, it is worth noting that our input images are feature fields with fibers that transform under the trivial representation of $SO(m)$, i.e., each RGB value in a color image stays invariant under rotation. The output of the encoder Enc (the pre-shapes) transform via *type-1* irreducible representations, which are just the rotation matrices $\rho_1(\mathbf{R}) = \mathbf{R}$. That is, a vector $z \in S_m^k$ of $k$ landmarks in $\mathbb{R}^m$ transforms via $k$ rotation matrices as specified in Subsec. 4.1. The `escnn` library further requires the specification for the hidden feature fields, for which we use regular representations (cyclic permutations), as these fields are compatible with usual activation functions such as ReLU.

## 5.2. The Encoder: Equivariant vMF Parameter Extraction

In the VAE setting, the encoder should encode for the parameters of vMF distribution on the $((k-1)m-1)$-dimensional hypersphere. The vMF is parametrized by a **mean** $\mu \in S_m^k$, i.e., $k$ landmarks with zero mean and unit norm. As such the encoder equivariantly obtains $\mu$ as $k$ type-1 features which we normalize to unit norm. During training we add a loss on the squared mean as to ensure that the encoder predicts centered landmarks. The vMF also requires a **concentration parameter** $\kappa \in \mathbb{R}_{>0}$, which the equivariant encoder predicts as a type-0 scalar $\tilde{k}$ that defines $\kappa$ via $\kappa = \text{softmax}(\tilde{\kappa}) + 1$ as to ensure a minimum variance of 1. Finally, the encoder also needs to predict a **pose** $\mathbf{R} \in SO(m)$. In 2D, this pose is encoded as a type-1 vector $\tilde{\mathbf{r}} \in \mathbb{R}^2$, which we normalize to $\mathbf{r} = \begin{pmatrix} r_x \\ r_y \end{pmatrix} = \frac{\tilde{\mathbf{r}}}{\|\tilde{\mathbf{r}}\|}$ from which we can extract a rotation angle via the arctan, or directly parametrize a rotation matrix $\mathbf{R} = \begin{pmatrix} r_x & -r_y \\ r_y & r_x \end{pmatrix}$.

## 5.3. The Posterior Shape-vMF is Invariant to Rotations

The three components (pose $\mathbf{R}$, shape $\mu$, concentration parameter $\kappa$) allow us to obtain a posterior distribution on the equivalence class of shapes $[z]$ by mapping the inferred $\mu$ to a $\mu_0$ which represents the canonical shape, or class representative, via $\mu_0 = \rho(\mathbf{R}^{-1})\mu$. It is noteworthy that the shape representative $\mu_0$ is invariant to input rotations, since due to equivariance of Enc we have that the parameters as a function of input image $f$ satisfy

$$\forall_{\tilde{\mathbf{R}} \in SO(m)}: \qquad \mu[\mathcal{L}_{\tilde{g}}f] = \rho(\tilde{\mathbf{R}})\mu[f] \quad \text{and} \quad \mathbf{R}[\mathcal{L}_{\tilde{g}}f] = \tilde{\mathbf{R}}\mathbf{R}[f].$$

Thus, vector representation of the equivalence class $[\mu_0]$ represented by $\mu_0$ is truly invariant:

$$\begin{aligned} \forall_{\tilde{\mathbf{R}} \in SO(m)}: \quad \mu_0[\mathcal{L}_{\tilde{g}}f] &= \rho(\mathbf{R}[\mathcal{L}_{\tilde{g}}f]^{-1})\mu[\mathcal{L}_{\tilde{g}}f] = \rho(\tilde{\mathbf{R}}\mathbf{R}[f])^{-1}\rho(\tilde{\mathbf{R}})\mu[f] \\ &= \rho(\mathbf{R}[f])^{-1}\rho(\tilde{\mathbf{R}})^{-1}\rho(\tilde{\mathbf{R}})\mu[f] = \rho(\mathbf{R}[f])^{-1}\mu[f] \\ &= \mu_0[f]. \end{aligned}$$

Give the above, we are thus able to learn a posterior distribution for the vector representative $z_0$, which represents the entire equivalence class $[f]$ via $q(z_0|f) = vMF(\mu_0, \kappa)$, since

$\forall_{f,f_0 \in [f]} : q(z_0|f) = q(z_0|f_0)$. In the generative mode, we sample from this distribution a representative latent $\hat{z}_0 \sim q(z_0|f)$, map it to the corresponding pose $\hat{z} = \rho(\mathbf{R}[f])z_0$ using the estimated pose $\mathbf{R}$, after which we pass $\hat{z}$ through the equivariant decoder to obtain the reconstructed image $\hat{f} = \mathrm{Dec}(\hat{z})$.

### 5.4. Summary: The KS-VAE recipe

- $\mathrm{Enc}(f) \to (\mathbf{R}, \mu, \kappa)$

- Extract the representative latent $z_0$, using $z_0 = \mathbf{R}^{-1}z$, where $z$ is latent with a canonical orientation.

- Sampling : $\hat{z}_0 \sim q(z_0|[f])$

- $\mathrm{Dec}(\hat{z}_0) \to$ transforms to $\hat{z}$ and then reconstructs the image $\hat{f}$.

## 6. Experiments

In this section, we perform experiments with KS-VAE and compare it with standard VAE and SVAE. Here all the models are trained in an unsupervised way, without any prior knowledge on the equivalence classes or the poses of every instance.

### 6.1. Qualitative results

For qualitative results, we demonstrate that KS-VAE is rotationally equivariant. For each equivalence class (or each orbit), KS-VAE can learn a representative $f_0$. The KS-VAE is able to encode every input image to its orbit, as well as extract the corresponding pose $R \in SO(m)$ that it has relative to the representative. We observe that the KS-VAE seems to map the digits 6 and 9 to the same equivalence class. This is efficient whilst still allowing to discern the two via the estimated pose within the class. For each image, it generates a *shape* which remains consistent when the image is transformed.

In Figure 2, the first column corresponds to a rotated digit from MNIST dataset, the second column is its reconstruction, the third column is the reconstruction using a canonical orientation and the last column contains a shape made using four landmark points for the different digits.

### 6.2. Quantitative Results

Here in Table 1, we quantify and characterise the differences in various VAE models in the task of equivariant learning and show that Shape-VAE outperforms standard VAE and SVAE.

All VAEs use the same architectures for the encoder and decoder; in KS-VAE, however, the conv-layers are replaced by their group convolutional counter parts for the $C_8$ cyclic permutation group (8 rotations covering $SO(2)$). The input images are padded with ones at the bottom and right in order to obtain images of size $29 \times 29$. The encoder performs 7 convolutions (followed by ReLU) with 5x5 kernels, without padding as to shrink the image to a single pixel. The decoder also consists of 7 layers with 5x5 convolutions, but with padding size 4 as to iteratively expand t he image size to $29 \times 29$. The feature channels

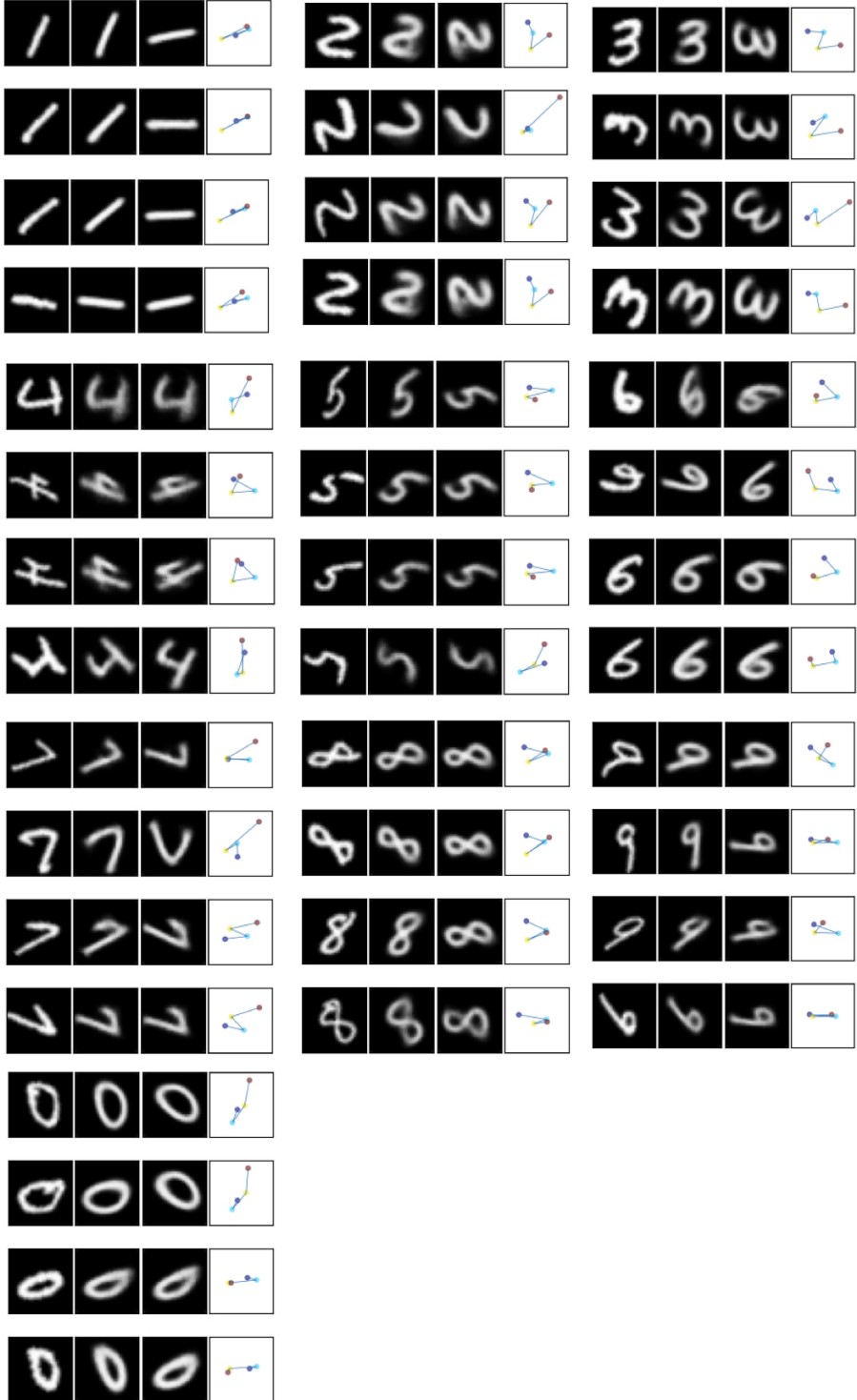

Figure 2: For each digit: Original rotated MNIST image, its reconstructions, reconstruction of the class representative ($f_0$ decoded from $z_0$), and inferred latent shape $z_0$ consisting of four landmarks.

increase (decrease) linearly in the encoder (decoder) from 8 in the first layer to 96 in the last layer.

Table 1: Comparing total loss (KL divergence + reconstruction error) for vanilla VAE, SVAE and KS-VAE, for training without rotated images and testing on rotated images. We note down the loss after training for 10 epochs to show the efficient learning in KS-VAE compared to VAE and SVAE

| Model | Model type | Dataset | Params | Loss |
|---|---|---|---|---|
| vanilla VAE | not equivariant | rot MNIST | 44.4M | 240.3 |
| SVAE | not equivariant | rot MNIST | 44.4M | 190.2 |
| KSVAE | equivariant | rot MNIST | 3.2M | 168.6 |

## 7. Discussion

Like any other hypserspherical model, this one too faces with the *vanishing surface area* problem. It shows unstable behavior of hyperspherical models in high dimensions. The surface area of a hypersphere for dimension $m$ is defined as

$$S^{m-1} = \frac{2\pi^{m/2}}{\Gamma(m/2)}.$$

As $S^{(m-1)} \to 0$ as $m \to \infty$, this leads to unstable behavior. Additionally different number of landmarks leads to different *shape* resulting to different shapes for the same class/image. However, this could be remedied using a conditional VAE and figuring out the optimal number of landmarks for the most intuitive *shape*.

It is noteworthy that the *shapes*, i.e., the collection of landmarks, do not exclusively contain information about shape in the common sense, but also includes variations in writing style and line thickness. In that sense, it is not any different to hypershperical VAEs except that our method is equivariant to rotations.

In future work, we intend to disentangle the representation in a pure shape component, and a separate appearance component which could encode for style, texture etc. As such, our method could provide the fundamentals for learning disentangled representations in terms of shape-pose akin to classical active appearance models.

## 8. Conclusion

In this paper, we proposed a novel VAE framework that successfully learns consistent symbols for equivalence classes of images, which disentangles pose from shape. In addition to the interpretable nature of the latent space, KS-VAEs outperform both standard VAEs as well as hyperspherical VAEs in terms of representational efficiency. This underpins the efficiency in representing abstract concepts in the form of geometric symbols. Towards this end, we presented a Kendall Shape-VAE framework for learning neural ideograms.

## Acknowledgments

Sharvaree Vadgama is funded by the Hybrid Intelligence Center, a 10-year programme funded by the Dutch Ministry of Education, Culture and Science through the Netherlands Organisation for Scientific Research. Erik J. Bekkers is financed by the research programme VENI (grant number 17290) funded by the Dutch Research Council. All authors sincerely thank everyone involved in funding this work.

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
