# OpenReview forum: "Kendall Shape-VAE : Learning Shapes in a Generative Framework"
_NeurIPS.cc/2022/Workshop/NeurReps — NeurReps 2022 Oral_

### Official Review · Reviewer_FnJa · 2022-10-08
**This is a well-written paper with relevant and robust contributions to geometric deep learning. I encourage authors to reformulate a few confusing or repetitive sections, as well as better visualize the model's architecture.**

**Confidence:** 4
**Soundness:** 4
**Presentation:** 3
**Contribution:** 4
**Overall Rating:** 8

**Summary:**

This paper presents the first VAE for equivariant and hyperspherical learning of shapes, rooted in a Kendall shape latent space, denoted Kendall Shape VAE (KS-VAE). The architecture is made up of an equivariant encoder/decoder combined with layers dedicated to rotation invariance. The training is unsupervised, with a typical ELBO + squared mean loss. The KS-VAE maps an equivalence class of translation/scaling (due to equivariant encoder) to a shape ("set of landmarks") which is then disentangled from the original image's orientation ("pose"). This "pose" is recorded as a rotation matrix such that the canonical latent variable ("class representative") can be rotated back to the original image's pose upon decoding. As such, the KS-VAE's latent space adopts a geometric structure that encodes shape in a rotation equivariant manner. In practice, this means that MNIST numbers like "6" and "9" share a similar canonical latent variable but can be re-separated during decoding using their respective poses. The authors further demonstrate how this geometric encoding enables more efficient learning compared to traditional VAE and spherical VAE models.

**Questions:**

- In the introduction section, what do you mean by tasks of domain generalization/data compression requiring "explicit control" over encoding? As in the encoding layers have some hard-coded properties that shape the latent space?

- Could this work be applied for encoding a shape's simplicial complexes and their connectivity? This could lead towards encoding a shapes' topological properties!

- Does the typical VAE enforcing data onto $S^1$ only fail for Gaussian distributions? (The sentence specifying this failure is confusing to read; I would encourage the authors to reformulate it in two smaller sentences).

- Does the approximate posterior $q_{\phi}(z|x)$ refer to the encoder? If so, it should be specified to match the " generator $p_{\theta}(x|z)$"

- The first sentence of the second paragraph of section 5.1 is confusing. What do the authors mean by RGB value being invariant under rotation? As in, the color does not change whether it belongs to a pixel at 0 rotation versus 90° rotation? If that is indeed the intended meaning, I would argue this is trivial information and can be omitted for clarity purposes.

**Limitations:**

Authors identify the limitations of the method with respect to the need for low dimensional data and entanglement of pure "shape" with style/handwriting of image.

It would be interesting to further quantify the latter limitation by taking the variance between canonical representations for a single MNIST digit class.

I would also appreciate the authors further specifying just how poorly the method scales with higher dimensional data. For example, how would the authors expect the hypersphere's surface area for m=3 to affect performance? While the authors specify higher dimensions would mean that there would be a greater choice of sets of landmarks for a single class, this does not provide much information into just how strict of a limitation this is.

**Recommended Decision:**

3: Accept

**Relevance:**

4: Highly relevant

**Strengths And Weaknesses:**

$\textbf{Originality: How novel and/or creative is the work? }$

This is the first work to the authors' knowledge to achieve rotation equivariant shape learning. The framework is innovative and makes creative use of geometric methods for more efficient and interpretable encoding.

$\textbf{Quality: Is the submission technically sound? Are claims well supported? }$

Yes, the authors provide a technically sound and well-supported mathematical model supporting their findings. MNIST results are clear, and performance metrics are convincing. I particularly liked the visual demonstration in Fig 2 of disentangled canonical representations in the latent space. It would be even more interesting if authors could quantify the variations between canonical representations of a single digit caused by change in style (e.x. take mean of representations and share standard deviation of that mean that would, in theory, be caused by variations in style).

I would like to see the full formulation of the loss function used during training. The ELBO is presented in section 3 and section 5.2 brings up a "loss on the squared mean," but the actual loss implementation is not explicitly given anywhere. It would be extra helpful to understand how this method might generalize to higher dimensions, even if it only be for 3D shapes.


$\textbf{Clarity: Is the submission well-organized and clearly written?}$

Clarity-wise:


The authors would greatly benefit from providing a better overview of the KS-VAE architecture. Figure 1 (which is currently not referenced anywhere in the text) should provide a clear and direct summary of section 5, such as a visualization of section 5.4, implementation-wise (where are the rotation invariant layers with respect to the translation/scaling equivariant layers?).  The sections 4 and 5 all independently address the encoder/decoder framework, but there is no clear overview of the implementation. Figure 1 should very clearly connect all the components, and reference sections 5.1, 5.2, and 5.4.

Organization-wise:


Sections 4.2 and 5.3 seem repetitive and could be combined into one succinct explanation of rotation equivariance in the context of Kendall shape space. If 4.2 presents the mathematical requirement for rotation equivariance, 5.3 could focus on the mere implementation of it (encoder mapping and generative model).

Writing-wise:
1. The KS-VAE acronym is not actually introduced anywhere. It first appears in caption fo Figure 1, and then in section 4.2. Moreover, Figure 1 is not referenced anywhere in the text.
2. There are some minor typos scattered throughout the paper that the authors would probably catch upon performing a thorough read-through. A few examples:
- "are mapped to an invariant shapes." should be "are mapped to invariant shapes";
- "network architectures, have multi-scale dependency" should be "network architectures, with multi-scale dependency";
- "to show a geometric symbol The" should be "to show a geometric symbol.) The";
- "(it's content stays" should be "(its content stays";
- "which representse" should be "which represents";
- "toz" should be "to z";
- "remains consistence" should be "remains consistent",
- "faces with the vanishing" should be "faces the vanishing";
- "Although this could be" should be "However, this could be";
- "learning disentangle representations" should be "learning disentangled representations";
- "learning neural ideograms" should be "learning neural ideograms."

$\textbf{Significance: Are the results important and of interest to this community?}$

These results are definitely of interest to the geometric learning community. Authors clearly explain their methodology and offer a significant contribution towards interpretable latent encoding, even if it is restricted to 2D data.


**Submission Track:**

Proceedings Paper (9 Page)

---

### Official Review · Reviewer_LSDd · 2022-10-15
**Review of Kendall Shape-VAE : Learning Shapes in a Generative Framework**

**Confidence:** 4
**Soundness:** 3
**Presentation:** 3
**Contribution:** 4
**Overall Rating:** 7

**Summary:**

In the introduction the authors cite extensions to VAEs that account for geometry, noting benefits. They introduce their approach with connects with Kendall shape spaces, noting the benefit of interpretability. Their focus is on what they call neural ideograms, a sort of non-arbitrary (natural) pictorial sign. They summarize their contributions, which include disentangling shape and pose in an equivariant VAE, whose hyperspherical latent space is a Kendall shape space.

In the related work section they introduce extensions to VAEs, including hierarchical VAEs, disentangling latent spaces, explaining latent spaces. They cite several works of equivariant VAEs, and distinguish their work from them by the use of a directional distribution versus Gaussian distributions on Euclidean space.

The background introduces the ELBO, the directional von Mises Fisher distribution for the hypersphere, and the analytical KL divergence between this distribution and its uniform prior.

In section 4, Kendall shape spaces are introduced.

In section 5 their implementation of their KS-VAE is detailed.

Section 6 details their experiments on rotated MNIST digits.

Section 7 they discuss the problem of vanishing surface area.

In the conclusion they emphasize the benefit of representational efficiency, supported by the lower loss and small parameters in Table 1.


**Questions:**

What is the value of m, and how does it connect with vanishing surface area? I thought it was nx*ny pixels, but then I thought it was 2 from the 2D rotation, and then I thought it was much larger because of the vanishing surface area. Please clarify.


**Limitations:**

1.
The wording of this sentence is unclear to me: “But this fails as in the low dimensions, Gaussian distribution presents probability mass around centered origin, while in high dimensions, it tends to resemble a uniform distribution on the surface of a hypersphere, with majority of its mass concentrated on the hyperspherical shell”

2.
In section “4.1. Kendall Shape Spaces” Does m would refer to: nx*ny for a grid of pixelated images of shape (nx,ny)? If so, could you explicitly state this. Also, I did not follow the reason for the last sentence "As a flattened vector..."

3.
In section “4.2. The Kendall Shape VAEs Requires Equivariant Encoders/Decoders” I am not sure what SIM is, so perhaps you could introduce it with a sentence and /or reference.

4.
In section “5.2. The Encoder: Equivariant vMF Parameter Extraction” you reference 2D. Does this mean that m=2? Also for “minimum variance of 1” I would report the approximate standard deviation value in degrees. You can take a large amount of samples and compute the empirical std.

5.
In section “5.4. Summary: The KS-VAE recipe” the last bullet point “transforms to z” needs a space between “to” and “z”

6. Typos etc

Typographical suggestion
it is worth the practical note that our input images → it is worth noting that practically speaking our input images

Typographical suggestion:
By approximating the posterior using vMF in SVAE, the resulting latents are separated
on the surface of the hypersphere as well as the model learns more efficiently
-->
By approximating the posterior using vMF in SVAE, the resulting latents are separated
on the surface of the hypersphere, and in addition the model learns more efficiently

Missing punctuation and bracket (figure 1 legend)
(We connect them with lines to show a geometric symbol The encoder maps both of these image from the same class to a canonical orientation and a set of landmarks


**Recommended Decision:**

3: Accept

**Relevance:**

4: Highly relevant

**Strengths And Weaknesses:**

Overall the disentanglement of shape and pose by their Kendall shape space is a simple and welcome contribution. I think it could have a much more general application beyond neural ideograms - e.g. all the domains where Kendall shape space analysis has been applied without deep learning.

The section "5.4. Summary: The KS-VAE recipe" was very helpful for clarify. However what the parameter m was remained ambiguous to me.

In figure 2 the inferred latent shapes did not seem to connect that clearly to the digits. Also, the figure wasn't annotated for expected results versus edge cases, so I wasn't sure if I was reading it correctly (comparing digits to see if the Kendall shape space was similar)

**Submission Track:**

Proceedings Paper (9 Page)

---

### Official Review · Reviewer_SZMw · 2022-10-17
**Extension of Hyperspherical VAEs to Kendall shape VAEs**

**Confidence:** 3
**Soundness:** 3
**Presentation:** 3
**Contribution:** 3
**Overall Rating:** 7

**Summary:**

This paper introduces Kendall Shape Variational Auto-encoders (KS VAE), unsupervised neural
networks that learn an encoding to a low-dimensional latent space and a decoder that
reconstructs inputs from latent codes. KS VAE enforces the latent space to represent the
Kendall shape space, that is the quotient of landmark configurations by the actions of scaling,
translation, and rotation. This relies on equivariant encoder and decoder networks, and the
use of probability distributions on the hypersphere. Their work improves over existing
Hyperspherical VAEs that only enforce invariance to scale and rotation.
First, a review of the literature is given, that distinguishes between equivariant VAEs
and Hyperspherical ones. This work lies at the intersection of both. Then some background on
VAEs and Hyperspherical VAEs (H VAE) is given and introduces the use of von Mises-Fisher
distributions to be used as posterior distribution. The KL divergence between uniform and
vMF distributions is tractable and given in closed form.
Kendall shape spaces and KS VAEs are then introduced. An image is first mapped
equivariantly to the pre-shape space via the encoder map that relies on previous work by
Knigge et al. and the escnn library. The obtained code is the parameter of a vMF distribution,
i.e. its mean. A concentration parameter and pose R (rotation matrix) are also computed by
the encoder. The authors then show that the obtained posterior distribution is invariant to
rotations. Finally, a decoder reconstructs the original image.
Experiments are then presented and show that the latent codes are indeed invariant
to rotation, and the value of the loss, on the rotated MNIST dataset. The loss is much
decreased after 10 epochs, while the number of parameters of the model is much reduced
compared to standard VAE and H VAE.
The limitations of the model are discussed in section 6 and perspectives are given.

**Questions:**

I don’t understand how a canonical representer of each class is obtained (as no global
section of Kendall shape space exist). Is the alignment/registration/Procrusted problem
solved to find the rotation that minimizes the distances between two preshapes at some
point?
Could this model be extended to 3d, knowing that sampling from vMF distribution is slightly
trickier in 3d?
Could spherical normal distributions (Hauberg 2018) be used instead of the vMF
distribution?
What kind of datasets/problem is this model most relevant for?
What would be the impact of this change?
Who is \tilde g in equations of section 5.3? Is it \tilde R?
How many samples contained the dataset?

**Limitations:**

The limitations of the proposed model are well discussed in section 6.

**Recommended Decision:**

3: Accept

**Relevance:**

3: Solid fit

**Strengths And Weaknesses:**

The paper is well written and clear and technically sound. Its originality lies in enforcing
rotation invariance in the latent space and modelling the latent space as a Kendall shape
space. This allows to reduce the number of parameters of the model and learn representations
of classes that are truly invariant. This invariance property is proved in section 5.3 and shown
by experimental results. However, the evaluation rather light as only one data set is used.
More experiments could show the relevance of the contribution.

**Submission Track:**

Proceedings Paper (9 Page)

---

### Decision · Program_Chairs · 2022-10-21

Accept (Oral)